# Exploring the Mechanism of Topical Application of Clematis Florida in the Treatment of Rheumatoid Arthritis through Network Pharmacology and Experimental Validation

**DOI:** 10.3390/ph17070914

**Published:** 2024-07-09

**Authors:** Ting Lei, Chang Jiang, Li Zhao, Jizhou Zhang, Qing Xiao, Yanhong Chen, Jie Zhang, Chunquan Zhou, Gong Wang, Jing Han

**Affiliations:** 1Institute of Materia Medica, Fujian Academy of Chinese Medical Science, Fuzhou 350003, China; 2220456047@fjtcm.edu.cn (T.L.); wanggong58@fjtcm.edu.cn (G.W.); 2Pharmacy College, Fujian University of Traditional Chinese Medicine, Fuzhou 350122, China

**Keywords:** Clematis Florida, rheumatoid arthritis, network pharmacology, NF-κB, MAPK

## Abstract

Clematis Florida (CF) is a folk medicinal herb in the southeast of China, which is traditionally used for treating osteoarticular diseases. However, the mechanism of its action remains unclear. The present study used network pharmacology and experimental validation to explore the mechanism of CF in the treatment of rheumatoid arthritis (RA). Liquid chromatography–mass spectrometry (LC-MS/MS) identified 50 main compounds of CF; then, their targets were obtained from TCMSP, ETCM, ITCM, and SwissTargetPrediction databases. RA disease-related targets were obtained from DisGeNET, OMIM, and GeneCards databases, and 99 overlapped targets were obtained using a Venn diagram. The protein–protein interaction network (PPI), the compound–target network (CT), and the compound–potential target genes–signaling pathways network (CPS) were constructed and analyzed. The results showed that the core compounds were screened as oleanolic acid, oleic acid, ferulic acid, caffeic acid, and syringic acid. The core therapeutic targets were predicted via network pharmacology analysis as PTGS2 (COX-2), MAPK1, NF-κB1, TNF, and RELA, which belong to the MAPK signaling pathway and NF-κB signaling pathway. The animal experiments indicated that topical application of CF showed significant anti-inflammatory activity in a mouse model of xylene-induced ear edema and had strong analgesic effect on acetic acid-induced writhing. Furthermore, in the rat model of adjuvant arthritis (AA), topical administration of CF was able to alleviate toe swelling and ameliorate joint damage. The elevated serum content levels of IL-6, COX-2, TNF-α, IL-1β, and RF caused by adjuvant arthritis were reduced by CF treatment. Western blotting tests showed that CF may regulate the ERK and NF-κB pathway. The results provide a new perspective for the topical application of CF for treatment of RA.

## 1. Introduction

RA is a chronic inflammatory disease primarily affecting the joints [1]. It is the most common form of inflammatory arthritis, characterized by chronic synovitis, synovial proliferation, and cellular infiltration. Furthermore, it leads to bone erosion, destruction of articular cartilage, intense joint pain, swelling, and a high rate of disability [2]. Current statistics indicate that RA affects approximately 0.5% to 1% of the global population and the disease affects women more than men [3,4]. The cause of RA is unclear and may be related to genetic, environmental, and immunological factors [5]. Conventional medicine primarily treats RA through non-steroidal anti-inflammatory drugs (NSAIDs), glucocorticoids, and disease-modifying antirheumatic drugs (DMARDs), including newer biologic agents [6]. These treatments aim to manage symptoms and slow disease progression but are not curative. Many anti-RA drugs have limitations such as poor bioavailability or gastrointestinal side effects, which decrease patient compliance. Topical routes of drug administration, which bypass the first-pass effect experienced with conventional oral administration, has emerged as a novel option, garnering increasing attention [7]. Topical drug administration provides rapid relief of localized swelling and pain, resulting in superior patient compliance [8].

Clematis Florida (CF) is a folk medicine in southeast China. It was recorded in the Chinese traditional medicine book *Zhong Hua Ben Cao* (中华本草) published in 1999, which describes its therapeutic effects of energizing the meridians, dispelling wind, activating blood circulation, and relieving pain [9]. Traditionally, CF is prepared by soaking in camellia oil and is applied topically to treat trauma, joint inflammation, rheumatic tendon pain, and other inflammatory conditions. Despite its centuries-long usage in folk practices, comprehensive scientific validation to support its efficacy remains scarce. CF belongs to the genus clematis in the Ranunculaceae family. Many plants of the genus clematis have a wide range of pharmacological effects, including relieving rheumatic pain, and treating cervical spondylopathy, scapulohumeral periarthritis, hepatic carcinoma, gastrointestinal conditions, etc. [10]. Notably, saponin-like compounds and triterpene saponins extracted from CF have demonstrated in vivo antitumor activity and the capacity to inhibit inflammatory mediators [11,12]. However, the specific compounds and the potential targets of CF for RA treatment still remain unclear. Moreover, the traditional usage of CF is for topical treatment of osteoarthritis-related diseases, but the pharmacological efficacy of topical CF application has not yet been reported. Such exploration not only bridges traditional knowledge and modern medical practice but also potentially expands the therapeutic arsenal against RA, offering hope for more effective and diverse treatment strategies.

The many compounds used in Chinese medicines make it complicated to explain the major and minor compounds as well as the main therapeutic targets for the treatment of diseases. The network pharmacology method makes it possible to elucidate the mechanism involved [13]. Network pharmacology combines pharmacological and systems biology approaches, which can be used to explain the mechanisms by which Chinese medicines treat diseases. The method emphasizes that the coordination of multiple signaling pathways can improve the therapeutic effect, from the traditional Chinese medicine (TCM) research model of “single drug, single target” to “multi-compound, multi-target, multi-path”, revealing the “drug–gene–target” relationship. This may be a good way to explain the mechanism of TCM in treating diseases [14]. In recent years, many studies have used network pharmacology to predict the interactions between compounds in TCM and target genes for specific diseases [15].

In this research, we used LC-MS/MS to detect and identify the chemical constituents of CF. Then, the network pharmacology method was used to predict potential targets and molecular signal pathways of CF in the treatment of RA, and was verified by subsequent animal experiments. These experiments aimed at providing essential experimental data for the potential therapeutic application of CF in RA treatment. The process flow of the study, detailing each step from chemical analysis to experimental verification, is clearly depicted in Figure 1.

## 2. Results

### 2.1. Main Compounds in CF Extract

To determine the composition of CF, the CF samples were analyzed by LC-MS/MS in both positive and negative ionization modes (Figure 2). In the base peak chromatogram (BPC) of the CF samples, a total of 50 compounds were identified (Appendix A). Then, 166 CF targets were identified by searching these compounds in the TCMSP, ETCM, and ITCM databases (Appendix A), and 2126 RA related targets were obtained from the DisGeNET, OMIM, and GeneCards databases (Appendix A). Finally, 99 overlapped targets were identified through the Venn diagram (Figure 3a). A total of 16 compounds that were reactive to the 99 overlapped targets were considered main compounds in CF for the treatment of RA.

### 2.2. Construction of CT Network and Screening Core Compounds in CF 

A compound–target (CT) network was constructed to visualize the interaction of CF compounds and candidate RA targets, using Cytoscape software (Figure 3b). It was revealed that 99 targets were regulated by 16 compounds in the CT network. Based on the CytoNCA analysis, the average degree value of all compounds was found to be 11.75. Compounds with degree values greater than average were considered core compounds (Appendix A). Five core compounds were screened, namely, oleic acid, oleanolic acid, ferulic acid, caffeic acid, and syringic acid.

### 2.3. Screening of Main Targets

The STRING database was used to analyze the KEGG pathway of overlapped targets (Figure 4b). All pathways were sorted according to false discovery rate (*FDR*) values. The top 30 pathways with lowest *FDR* values are listed (Appendix A). From this list, 9 of the top 30 pathways that showed the highest correlation with inflammation were screened (Table 1). The corresponding targets in these pathways were predicted as the main targets of CF in the treatment of RA.

### 2.4. GO Enrichment Analysis

According to the gene ontology (GO) enrichment analysis, the top 10 results with the smallest *p*-values for biological processes (BPs), cellular components (CCs) and molecular function (MF) were listed (Figure 4a). As a result, the most prominent targets identified via GO included stress-activated MAPK cascade and I-kappaB/NF-kappaB complex. 

### 2.5. Construction of CPS Network and Screening of Core Targets

Cytoscape was used to construct a compound–potential target genes–signaling pathways (CPS) network (Figure 3c). Utilizing the Cyto-NCA tool within Cytoscape, we predicted the core targets of CF. Targets were ranked based on their degree value from the highest to the lowest (Table 2). Furthermore, targets with a degree value three times higher than the average were screened as the core targets. These core targets were PTGS2, MAPK1, NFκB1, TNF, and RELA. 

In total, 1568 targets related to RA were obtained from the DisGeNET database, and these targets were used to construct an RA-related gene network in the STRING database. The network was analyzed using Cytoscape. Results showed that the RA-related gene network consisted of 552 nodes and 10,865 edges. Then, the core targets obtained from the above CPS network were screened and ranked by degree value (Table 3). These targets ranked high in degree within the RA-related gene network, suggesting their significance as key therapeutic targets in the pathogenesis of the disease. 

### 2.6. Molecular Docking Effectiveness

Molecular docking was performed to validate the interaction between core compounds of CF (oleanolic acid, syringic acid, oleic acid, caffeic acid, and ferulic acid) and the core targets predicted through the CPS network (MAKP1, NFκB1, TNF-α, and PTGS2) (Figure 5). As a positive control, known inhibitors of NFKB1, MAPK1, and COX-2 (dexamethasone [16], ulixertinib [17], and rofecoxib [18], respectively) were assessed for their binding sites and binding energies with the targets, using molecular docking. Compared with the known inhibitors, the CF core compounds were found to bind to the same site on the targets (Table 4). Among all the compounds, oleanolic acid had binding energies lower than −7 kcal/mol with MAPK1 and PTGS2. The free binding energies between the other compounds and their corresponding targets were relatively low (Table 4). Thus, these core compounds of CF may play a significant role in RA treatment by targeting these core targets.

### 2.7. HPLC Measurement of Quality Control Substance Oleanolic Acid Content

The HPLC results indicated that the oleanolic acid content in the CF was 1.34% (Figure 6). This quantification is crucial because oleanolic acid is one of the key bioactive compounds contributing to the therapeutic properties of CF.

### 2.8. CF Alleviated Xylene-Induced Mouse Ear Edema

The anti-inflammatory effect of CF was assessed using xylene-induced mouse ear edema. As shown in Figure 7a, xylene application induced significant ear edema. After the intervention of CF-H administration, the degree of ear edema significantly decreased and the effects were dose-dependent (*p* < 0.001).

### 2.9. CF Showed Analgesic Activity in Acetic Acid-Induced Writhing Model

In the treatment of RA, it is important for drugs to exert analgesic effects to alleviate pain and stiffness. Acetic acid-induced writhing tests were performed to examine the analgesic activity of CF (Figure 7b). In the acetic acid-induced writhing test, CF-H treatment significantly decreased the number of writhes (*p* < 0.001). The effects were also shown to be dose-dependent. The data indicated that CF was effective in ameliorating the pain induced by chemical stimulation.

### 2.10. CF Alleviated the Severity of Arthritis in Rats

Figure 8a,b show the effect of CF on toe swelling in the adjuvant arthritis (AA) model rats. Animals in the model group showed significant toe swelling after injection of Freund’s adjuvant (*p* < 0.001). The degree of toe swelling in the CF-H group was significantly reduced compared with the model group when the treatment continued for 14 days (*p* < 0.01). There was no significant difference in the effect of reducing toe swelling between the CH-H and DD groups. The CF-L group showed no efficacy in reducing toe swelling, indicating a dose-dependent response to the treatment. The results indicated the efficacy of the CF-H treatment in alleviating RA symptoms. During the experiment, no significant differences in the body weights of the animals in different groups were observed (Figure 8c), indicating that the test did not affect the general condition of the rats. This result may have been due to the fact that the administration route was topical, which had little influence on the general condition of rats.

### 2.11. Effects of CF Extracts on Serum IL-6, COX-2, TNF-α, IL-1β, and RF in RA Rats

Figure 9a–e presents the serum levels of inflammatory factors in RA rats. In the RA model group, there were significant increases in the serum levels of IL-6, COX-2, TNF-α, RF, and IL-1β compared with the control group (*p* < 0.01). The CF-H groups showed significant reductions in the serum levels of the inflammatory factors (*p* < 0.01). As positive control, DD treatment was able to significantly reduce the levels of RF, COX-2, TNF-α, and IL-1β (*p* < 0.05). Compared with the DD group, CF-H had a more significant effect on reducing the serum COX-2 level (*p* < 0.001). As for IL-6 and TNF-α, the reduction effects in the CF-H group were better than in the DD group, although there was no significant comparative difference. CF-L groups exhibited notably lower serum levels of COX-2, TNF-α, IL-6, and IL-1β compared with the model group (*p* < 0.05, *p* < 0.01). The results indicated that CF was able to reduce the level of serum inflammatory factors in a dose-dependent manner, and the effect of CF-H group was even better than that of the positive control. These results indicated that CF may improve RA by influencing these targets.

### 2.12. Protein Expression of p-ERK, ERK, p-NF-κB and NF-κB in Rats

Compared with the control group, the expression of p-ERK/ERK in the joint tissues of rats in the model group was significantly reduced (*p* < 0.01), and the expression of p-NF-κB/NF-κB was significantly increased (*p* < 0.001). Compared with the model group, the expression of p-ERK/ERK showed an increase and the expression of p-NF-κB/NF-κB showed a decrease in the CF-H group (*p* < 0.05). These results indicated that CF was able to modulate the expression of proteins in the ERK and NF-κB signaling pathways, thereby exerting its anti-RA effects (Figure 9f–h).

### 2.13. Histological Analysis of Rats 

The ankle joint tissues of rats in the control group showed no signs of inflammation or damage. The synovial tissue was structurally intact, with no inflammatory infiltration, indicative of normal joint health. The rats in the model group exhibited significant pathological changes. A large number of inflammatory cells infiltrated the joint cavity, and the articular cartilage structure was notably degraded. In the groups treated with CF at different dosages, a significant improvement was observed in joint tissue health. There was only a small amount of inflammatory cell infiltration in the ankle joints, and the structure of the articular cartilage remained largely intact. Notably, the CF-H treatment group showed a more pronounced improvement compared with the CF-L group, suggesting a dose-dependent effect of the treatment. In the group treated with DD, the infiltration of inflammatory cells and joint destruction were reduced compared with the model group (Figure 10). These findings suggest that CF has a therapeutic effect on RA. 

## 3. Discussion

In this study, an integrated strategy combining LC-MS/MS and network pharmacology was used to elucidate possible effective compounds and the core therapeutic targets of CF in its anti-RA effects. Then, animal experiments were conducted to observe the therapeutic effects of topical administration of CF on RA and to verify the targets screened by network pharmacology. Our results showed that topical application of CF could mitigate inflammation, show analgesic activity, improve the symptoms of RA model rats, and reduce the levels of serum inflammatory factors IL-6, COX-2, TNF-α, IL-1β, and RF. Moreover, CF had significant effects on NF-κB and ERK MAPK pathways in joint tissue. The results validated the findings of the network pharmacology analysis.

There have been no previous reports on the analysis of the complete composition of CF using LC-MS/MS methods. In the present study, LC-MS/MS revealed that the CF contained a rich variety of chemical compounds. Through the analysis of the CT network, the core compounds of CF in its anti-RA action were predicted as oleanolic acid, oleic acid, ferulic acid, caffeic acid, and syringic acid. All these compounds have been reported to possess anti-inflammatory bioactivity. Oleanolic acid, a pentacyclic triterpenoid compound, is widely distributed in nature and exhibits various biological activities including anti-tumor, anti-inflammatory, antiviral, and antioxidant properties [19]. The literature indicates that CF is rich in saponins, primarily oleanane-type triterpene saponins [20]. Oleic acid, a monounsaturated fatty acid, is known for its anti-inflammatory and antioxidant effects [21]. Ferulic acid exerts a variety of biological activities, especially relating to oxidative stress, inflammation, vascular endothelial injury, fibrosis, apoptosis, and platelet aggregation [22]. Caffeic acid is a widely distributed hydroxycinnamic acid salt and phenylpropanoid metabolite in plant tissues, with multiple biological activities such as antioxidant, anti-cancer, antiviral, anti-inflammatory, and anti-diabetic effects [22,23,24,25]. Syringic acid can inhibit the release of inflammatory mediators and alleviate inflammation, and possesses additional properties such as antioxidant capability [26,27]. 

It was worth noticing that in the molecular docking results, only the binding degree of oleanolic acid with MAPK1 and PTGS2 was lower than −7 kcal/mol, which was considered to be the threshold of significant binding energy. Meanwhile, the binding energies between the other compounds and their corresponding targets were higher than this threshold. As we all know, traditional Chinese medicine often contains a large number of compounds, and the effect of multiple components on multiple targets is a significant feature of the efficacy of traditional Chinese medicine. Chinese medicine may contain a variety of components that can act on a single target; a single component may not have the ideal binding energy, but when multiple components work together, they may produce a significant effect on the target. In the present study, the core compounds oleanolic acid, caffeic acid, ferulic acid, syringic acid, and oleic acid were all able to bind to a common target, PTGS2. Although not every component had a binding energy lower than −7 kcal/mol with PTGS2, they may have been able to work together on the target to produce an effect. This phenomenon is common in traditional Chinese medicine research. Many research studies have reported weak binding between traditional Chinese medicine components and their targets [28]. However, traditional Chinese medicine can still achieve the ultimate therapeutic effect, attributed to the synergistic interactions between each individual compound [29]. In the present study, it was found that the ethanol extract of CF applied topically had significant anti-inflammatory and analgesic activities and had significant pharmacological effects against RA. These effects may have been the result of a synergistic action of multiple compounds.

Meanwhile, network pharmacology revealed that 99 potential targets of CF were responsible for the RA treatment. Among them, PTGS2, MAPK1, NF-κB1, TNF, and IL6 were the core targets, which was experimental validated. PTGS2 (prostaglandin G/H synthase 2), also known as COX-2, significantly promotes prostaglandin production in synovial tissue of RA patients. It is one of the common targets for the treatment of RA [30]. Along with IL-6 and TNF-α, it is one of the inflammatory factors that play a key role in the pathogenesis of RA. When the expression of COX-2 is inhibited, feelings of pain can be relieved [31]. IL-6, a member of the pro-inflammatory cytokine family, is an important inflammatory mediator in the RA disease process [32]. Excessive and sustained dysregulation of IL-6 synthesis can have pathological effects on chronic immune-mediated diseases. TNF plays crucial roles in various cellular processes, including apoptosis, cell survival, and immune regulation [33]. TNF-α directly affects osteocyte RANKL expression and increases osteoclastogenesis [34]. 

NF-κB is the core activated protein in a wide range of autoimmune diseases, including RA [35]. Activated NF-κB can participate in the inflammatory response by regulating cytokines, adhesion molecules, and chemokines. The release of activated NF-κB into the nucleus also induces the downstream production of pro-inflammatory cytokines such as COX-2, TNF-α, IL-6, and IL-1β, further exacerbating the inflammatory response [36]. In our study, ELISA results showed that the serum COX-2, TNF-α, and IL-1β levels in RA rats were significantly elevated, which was consistent with other reports [37,38]. After the administration of CF intervention, their levels decreased significantly. Meanwhile, the activation of NF-κB p65 was inhibited by CF, indicating that the activation of the NF-κB signaling pathway in rat synovial tissue was effectively inhibited. As mentioned above, COX-2, TNF-α, IL-6, IL-1β, and RF may be induced by the activation of NF-κB, so NF-κB was verified as one of the key targets for CF to treat RA.

MAPK1 (mitogen-activated protein kinase 1), also known as ERK1 (extracellular signal-regulated kinase 1), is a pivotal member of MAPK family. The MAPK signaling pathway plays a role in the cellular function of synoviocytes, chondrocytes, and bone marrow mesenchymal stem cells in knee joints [39]. Studies have shown that activating the MAPK pathway may regulate the secretory metabolism of cartilage, thereby promoting the production of extracellular matrix by chondrocytes and protecting cartilage [40]. The phosphorylation and activation of ERK1/2 and Akt may upregulate the expression of cell cycle proteins in MSCs35 [41]. Activation of the ERK MAPK pathway could suppress the autophagy of chondrocytes and promote the proliferation of chondrocytes, since the ERK MAPK pathway is associated with multiple growth factor signaling pathways that regulate cell proliferation and tissue homeostasis [42]. In the present study, we found that the expression of p-ERK1/2 protein in CFA-induced arthritis rats was elevated by CF treatment, suggesting that CF may have promoted the proliferation of chondrocytes by targeting the ERK1/2 signaling pathway, which may have provided protection for chondrocytes.

TCM has shown distinct advantages in treating RA, with ethnobotanical medicinal plants being an important source for developing anti-RA drugs [43]. CF is a plant in the Ranunculaceae family that is primarily known for its anti-inflammatory and analgesic effects. Active ingredients isolated from the Clematis genus are frequently used to treat RA, boost immunity, and as an adjunct therapy for cancer [44,45]. However, current research indicates that Clematis genus drugs possess certain toxicity [10]. Preliminary studies by our research group have revealed that the oral median lethal dose (LD50) of CF in mice is 60.08 g/kg, indicating potential toxicity at high oral dosages [46]. Nevertheless, our previous studies have also shown that topical application of CF does not elicit toxic responses [47]. Topical administration of TCM in treating RA offers multiple advantages, such as reducing the side effects associated with oral administration, avoiding the first-pass effect in the gastrointestinal tract, ease of use, and the ability to immediately discontinue treatment if adverse reactions occur [48]. To sum up, CF is a safe and effective topical drug in the treatment of RA.

## 4. Materials and Methods

### 4.1. LC-MS/MS Analysis

#### 4.1.1. Sample Preparation

The samples of CF were sourced from the Min Dong She Green Herb Development Association (Ningdei, China). The samples were identified by Zehao Huang from Fujian University of Traditional Chinese Medicine and stored in the Institute of Materia Medica, Fujian Academy of Chinese Medical Science. For preparation, 20 mg samples of CF powder and 300 µL of 40% methanol solution were mixed using the vortex mixing method. Every mixture was centrifuged at 16,000× *g* (4 °C) for 15 min. The resulting supernatant was collected for analysis. 

#### 4.1.2. Instruments and Analytical Conditions

The chemical composition of CF was detected and scanned using a UHPLC-Q Exactive Focus mass spectrometer (Thermo Scientific, Bremen, Germany) equipped with an ESI source. The mobile phase consisted of 0.1% methanol in water (phase A) and 0.1% methane in acetonitrile (phase B). The gradient elution scheme was as follows: 0–17.0 min, 5–98% B; 17.0–17.2 min, 98–5% B; 17.2–20.0 min, 5% B. The time-of-flight mass range was 90–1300, the spray voltage was 3800 volts for ESI+ and 3500 volts for ESI-, the sheath gas flow rate was 40 L/min, the ion transfer tube temperature was set at 320 °C, and the nebulization temperature was maintained at 350 °C. MS data were acquired using the mass spectrometer and processed with ProteoWizard 4.7.2 software. 

#### 4.1.3. Screening of Main Compounds of CF

Compounds exhibiting high abundance values were selected from the mass spectra. Additionally, the relevant literature was reviewed to supplement our understanding of the main constituents of CF. This comprehensive approach enabled us to finalize the main constituents of CF. 

### 4.2. Network Pharmacology Analysis

#### 4.2.1. Candidate Therapeutic Targets of CF Screening

The TCMSP database (https://old.tcmsp-e.com/tcmsp.php, accessed on 10 July 2023), the ETCM database (http://www.tcmip.cn/ETCM/index.php, accessed on 12 July 2023), and the ITCM database (http://itcm.biotcm.net, accessed on 12 July 2023) were utilized to predict all targets associated with the main compounds identified by mass spectrometry. SwissTargetPrediction (http://www.swisstargetprediction.ch/result.php, accessed on 15 July 2023) was employed to predict the compounds that lacked target information, with a focus on targets with a confidence level of 0.8 or higher. These targets were then standardized into gene names, using the UniProt database (https://www.uniprot.org/, accessed on 20 July 2023). Ultimately, the prediction results from all these databases were compiled and organized.

#### 4.2.2. Construction of RA-Related Target Database

The DisGeNET database (https://www.disgenet.org/, accessed on 10 September 2023), the OMIM database (https://www.omim.org/, accessed on 10 September 2023), and the GeneCards database (https://www.genecards.org/, accessed on 10 September 2023) were accessed to screen targets related to “arthritis”. Subsequently, the data obtained from these sources were merged and converted into standard gene names in Uniprot.

#### 4.2.3. Intersection between Main Compounds and Disease Targets

The online tool Venny 2.1 (https://bioinngp.cnb.csic.es/tools/venny/index.html, accessed on 15 September 2023) was employed to intersect the disease targets with the drug targets to identify the possible targets of CF against RA. 

#### 4.2.4. Construction of CT Network 

The STRING database (https://cn.string-db.org, accessed on 25 September 2023) was used to analyze the CF and RA targets that overlapped. The organism was set to Homo sapiens and the minimum required interaction score was >0.7. Cytoscape 3.7.1 software was used for network topology analysis. Subsequently, the core compounds within this network were analyzed using the CytoNCA 2.1.6 plugin in Cytoscape.

#### 4.2.5. CPS Network Construction and Core Target Acquisition

STRING was used to perform KEGG pathway enrichment analysis. We prioritized pathways based on their FDR, from lowest to highest. Among the top 30 pathways, 9 pathways that were related to inflammation were selected for primary analysis. The targets within these 9 inflammation-related pathways, which are directly involved in treating RA, were designated as the primary targets. They were imported into Cytoscape for the construction of the CPS network. Core targets were screened through the CytoNCA plugin for Cytoscape. The DisGeNET database was used to collect targets related to RA. These targets were imported into STRING to construct an RA-related gene network, which was analyzed using the CytoNCA 2.1.6 plugin in Cytoscape. The core targets were screened and ranked based on their degree value to confirm their importance in the pathogenesis of diseases.

#### 4.2.6. GO Enrichment Analysis

The predicted targets were analyzed for GO enrichment using R 4.4.4 software (https://www.r-project.org, accessed on 28 September 2023). ClusterProfiler, GOplot, and Pathview are the major visualization packages included in the R package. 

#### 4.2.7. Molecular Docking

Molecular docking was conducted to validate the interaction between core compounds and their corresponding targets. As a positive control, the known inhibitors of the targets of interest were also selected to perform molecular docking. The SDF format files of the core compounds were downloaded from the PubChem database (https://pubchem.ncbi.nlm.nih.gov, accessed on 10 December 2023) and converted into MOL2 format files. The PDB structure files of the target proteins were acquired from the RCSB PDB database (https://www.rcsb.org, accessed on 11 December 2023). These targets were prepared by removing water molecules and adding hydrogen atoms using AutoDock Tools 1.5.7. Molecular docking simulations were performed with AutoDock Vina 1.2.3, and the results were visualized using PyMOL 2.5.7 software.

### 4.3. Quality Control of CF Extract

According to reports, CF is rich in saponins, primarily oleanane-type triterpene saponins [20], and oleanolic acid has commonly been used as a quality control substance for medical clematis plants [49]. So, oleanolic acid was determined as a quality control substance for CF extract using the HPLC method.

#### 4.3.1. Sample and Standard Solution Preparation

A sample of 10 g CF powder was dissolved in 150 mL methanol by ultrasonication for 70 min, then filtered using a 0.45 µm membrane filter. A standard solution of oleanolic acid was prepared by accurately weighing 5.0 mg of the oleanolic acid standard. This oleanolic acid standard was dissolved in 5.0 mL of methanol to create the stock solution. 

#### 4.3.2. Instruments and Analytical Conditions

HPLC analysis was carried out using a Leaps UHPLC system (Chromai, Beijing, China). The chromatography was performed on a Hypersil ODS C18 column (250 mm × 4.6 mm, 5 µm). Water (10% A) and acetonitrile (90% B) were used as mobile phase solvents, utilizing an isocratic elution system. The mobile phase flow rate was kept at 1 mL/min. The column temperature was kept constant at 35 °C, and the samples were analyzed at 205 nm. The final content of the analytes was quantified using the external standard method.

### 4.4. Experimental Validation

#### 4.4.1. Animals 

All animals were purchased from Beijing Huafukang Bio-technology Co. (Beijing, China). Animals were kept under conventional laboratory conditions. All animals were fed water and standard maintenance food ad libitum. The acclimation period for animals was one week. Animal experimental procedures and methods were approved by the Animal Ethics Committee of Fujian Academy of Traditional Chinese Medicine on March 10, 2023 (FJATCM-IAEC2023024). 

#### 4.4.2. Preparation of the CF Extract

A sample of 1000 g CF powder was crushed and subjected to extraction with 70% ethanol at 90 °C, performed twice. The extract was subsequently concentrated under reduced pressure at 70 °C to obtain 298 g extract. The extract was diluted with water to 0.8 g/mL as a high-dose CF sample and 0.4 g/mL as a low-dose CF sample. This extract was prepared and reserved for further use.

#### 4.4.3. Xylene-Induced Mouse Ear Edema

A total of 32 ICR male mice (18–22 g, 7–9 weeks) were randomly divided into 4 groups with 8 mice in each group, which included the xylene model group (0.9% normal saline), positive control group (diethylamine diclofenate, DD), high-dose CF group (0.8 g/mL, CF-H), and low-dose CF group (0.4 g/mL, CF-L). For this experiment, 20 μL of CF was applied on both sides of the right ear in the CF-H and CF-L mice, and an equal amount of normal saline was applied to the mice in the model group. Mice in the positive control group were had DD applied at a thickness of 1 mm. The administration was performed once every 10 min for a total of 6 times. Then, 10 min after the last administration, 40 μL of xylene was applied to the right ear, whereas the left ear was left untreated. After 30 min, the mice were sacrificed. The ears were collected using a biopsy punch with a diameter of 6 mm and weighed. Ear edema was defined as the difference in weight between the right ear and the left ear.

#### 4.4.4. Acetic Acid-Induced Abdominal Writhing Response

Another 32 ICR male mice (18–22 g, 7–9 weeks) were randomly divided into 4 groups with 8 mice in each group, which included the model group (0.9% normal saline), positive control group (diethylamine diclofenate, DD), high-dose CF group (CF-H), and low-dose CF group (CF-L). One day before the experiment, the hair covering on the abdomen of each mouse was removed with 8% sodium sulfide. On the second day after hair removal, mice in the CF-H and CF-L groups received topical application of 0.05 mL CF on the abdomen. The mice in the model group were treated with the same dose of normal saline, while the positive control group was treated with 0.05 g of DD. The administration was performed once daily for 5 days. On the 5th day, the drugs were administrated twice with a 30 min interval. Then, 1 h after the last administration, the mice were intraperitoneally injected with 0.2 mL of 0.6% acetic acid per mouse. A writhe was defined as the contraction of the abdomen and pelvic rotation, followed by the extension of the hind limbs. The number of writhes in each group of mice within 5–20 min after injection of acetic acid was observed and recorded. 

#### 4.4.5. AA Model Preparation

Forty SD rats (half male and half female, 160–260 g, 6–8 weeks) were securely positioned in a supine posture within an immobilizer. Subsequently, 0.1 mL of Freunds Complete Adjuvant (FCA, Beyotime, Shanghai, China) was injected intradermally into the dorsum of the left hind toe. One minute following the injection, the rats were returned to their cages.

#### 4.4.6. Experimental Grouping and Drug Administration

The rats were divided into 5 groups with 8 rats in each group, which included the control group (0.9% normal saline), model group (0.9% normal saline), positive control group (diethylamine diclofenate, DD), high-dose CF group (CF-H), and low-dose CF group (CF-L). All groups except the normal group received an injection of 0.1 mL of FCA in the left hind toe, to construct the AA model. From the second day after inflammation onset, the CF treatment groups received topical applications of CF-H (0.8 g/mL) and CF-L (0.4 g/mL) doses to the affected toes daily for 28 consecutive days. The positive control group received a daily topical application of 0.2 g DD (10 mg/g, Novartis Pharma, Beijing, China), continuing for 28 days. The model control group and the normal control group were treated with 0.9% normal saline applied topically. 

#### 4.4.7. Detecting the Degree of Toe Swelling in Rats

Foot volumes and body weights of rats were measured on the day before the AA model was assembled, and 7, 14, 21, and 28 days after modeling. The foot volume of rats was assessed in a toe volume measurement machine (KEW, Shanghai, China). A total of 5 measurements were taken. The degree of toe swelling for each group of rats was then calculated. The degree of toe swelling was calculated as (toe volume after modeling-toe volume before modeling)/toe volume before modeling × 100%.

#### 4.4.8. Determination of Serum Inflammatory Factor

After the final toe volume measurement, the rats were euthanized under sodium pentobarbital anesthesia. Subsequently, 5 mL of blood was collected from the abdominal aorta. The blood was then centrifuged at 3000× *g* for 10 min to separate the serum. The contents of COX-2, TNF-α, IL-6, IL-1β, and RF in the serum were determined using ELISA kits (Elabscience, Wuhan, China, E-EL-R0792c, E-EL-R2856c, E-EL-R2856c, E-EL-M0037; Mlbio, Shanghai, China, ML-EA-A02702). 

#### 4.4.9. Hematoxylin and Eosin Staining

Ankle joint tissues were harvested and fixed in 4% paraformaldehyde, followed by decalcification with 10% neutral buffered EDTA. The tissues were obtained, fixed, decalcified, embedded in paraffin, and sectioned at a thickness of 5 µm. The joint sections underwent deparaffinization using xylene and were dehydrated in a gradient ethanol series. The sections were stained with hematoxylin for 5 min, differentiated with 1% hydrochloric acid ethanol for 30 s, treated with 0.2% ammonia for bluing, and stained with 0.5% eosin for 10 min. Finally, the sections were observed under a light microscope.

#### 4.4.10. Western Blotting

Joint tissue was obtained after the final toe volume measurement. Total protein was extracted using a column tissue protein extraction kit (Yaenzyme, Shanghai, China). For every 10 mg of tissue, 100 μL of lysis buffer was added. The mixture was thoroughly ground using the Servicebio high-speed cryogenic tissue grinder (Servicebio, Wuhan, China). The lysate was transferred to an ultrafiltration centrifuge tube and centrifuged at 12,000× *g* (4 °C) for 2 min. The protein concentration was determined using a BCA protein assay kit (Yaenzyme, Shanghai, China). Then, 15 μg of protein sample was subjected to sodium dodecyl sulfate–polyacrylamide gel electrophoresis (SDS-PAGE) and then transferred to a 0.45 μm PVDF membrane (Yaenzyme, China). The gel blocking was performed with blocking solution for 30 min, followed by incubation with primary antibodies overnight at 4 °C: NF-κB p65 (1:4000, Cell Signaling Technology, Boston, MA, USA, Rabbit mAb, #3033S), NF-κB (1:1000, BIOSS, Mouse mAb, bsm-33117M), ERK p44/42 (1:2000, ABclonal, Wuhan, China, Rabbit mAb, A4782), ERK (1:1000, Cell Signaling Technology, Rabbit mAb, #4695S). After washing three times with TBST, the membrane was incubated with HRP-conjugated goat anti-rabbit secondary antibody (1:5000, Cell Signaling Technology, #7074S) or HRP-conjugated goat anti-mouse secondary antibody (1:10,000, BIOSS, Beijing, China, bs-0296G-HRP) at 26 °C for 2 h. After washing three times with TBST, the proteins were made visible through ECL chemiluminescent substrate (Yaenzyme, China) and the intensities were measured using ImageJ 13.0.6 software (n = 5).

### 4.5. Statistical Analysis

Data were expressed as mean ± SEM and were analyzed by one-way analysis of variance (ANOVA) when the series of each group were normally distributed. The LSD test was used to compare between groups when the variance was the same; Games–Howell was used to compare between groups when the variance was not the same. The significance level was *p* < 0.05. All statistical analyses were performed using SPSS 20.0 software.

## 5. Conclusions

In this study, LC-MS/MS and network pharmacology were used to screen and predict the core compounds of CF for treating RA. These compounds were identified as oleanolic acid, oleic acid, ferulic acid, caffeic acid, and syringic acid. The key targets were screened and found to be PTGS2 (COX-2), MAPK1 (ERK1), IL6, TNF, and RELA, primarily involving the ERK MAPK pathway and NF-κB pathway. The pharmacological effects and mechanisms of the topical application of CF in treating RA were also validated in animal experiments. The results of this study provide theoretical support for the use of CF in the topical treatment of RA and establish a theoretical basis for its use. The present study demonstrates that CF is a new and potentially valuable natural drug for the topical treatment of RA.

## Figures and Tables

**Figure 1 pharmaceuticals-17-00914-f001:**
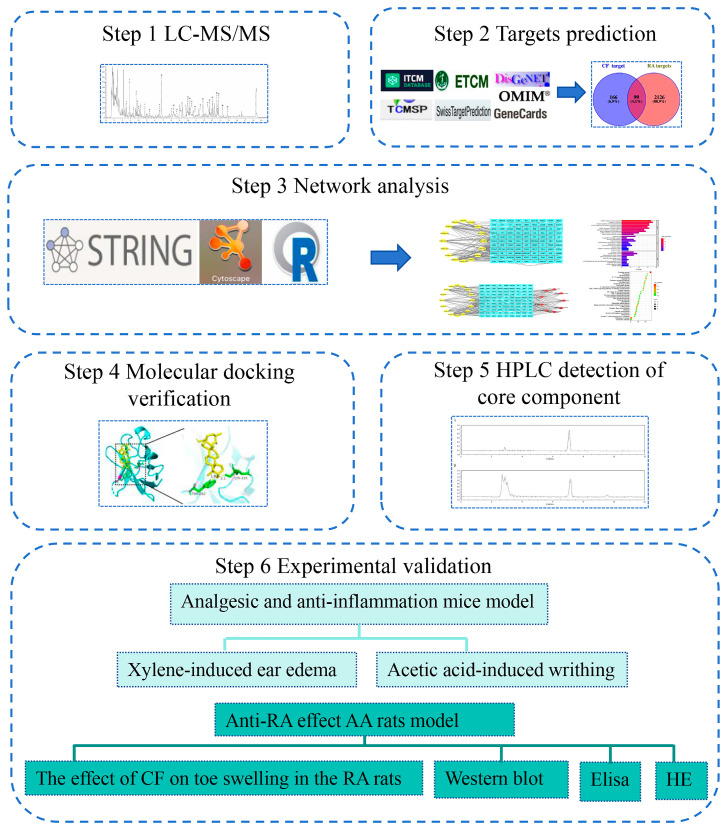
Flowchart of the methodology for studying the mechanism of action of topical application of CF in the treatment of RA.

**Figure 2 pharmaceuticals-17-00914-f002:**
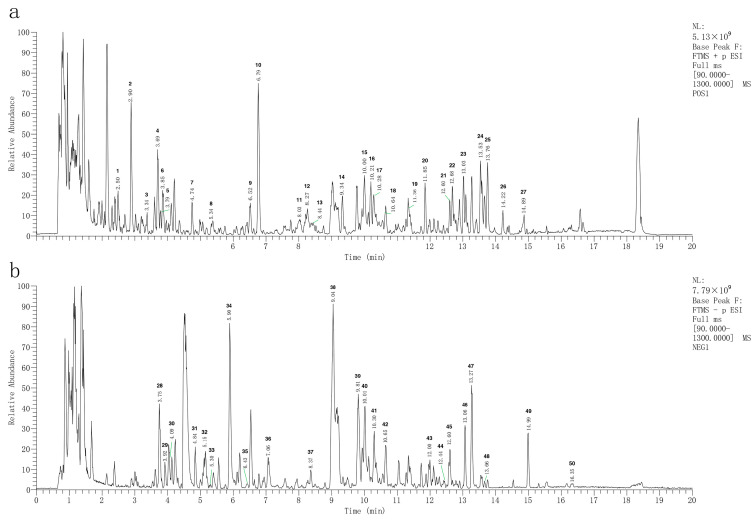
Total ion current diagram of CF extraction solution: (**a**) positive ion detection mode; (**b**) negative ion detection mode.

**Figure 3 pharmaceuticals-17-00914-f003:**
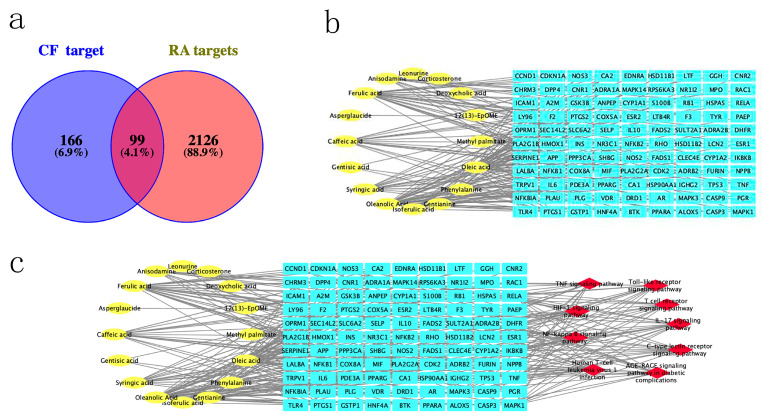
Network pharmacological analysis of CF for treating RAL: (**a**) Venn diagram showing intersection of CF-related targets and RA-related predicted targets; (**b**) plot of association between CF compounds and RA. The 16 yellow nodes on the left represent the main compounds of CF, and the 99 CF targets for the treatment of RA are labeled in blue on the right-hand side; (**c**) CTS network of CF in treatment of RA. Yellow nodes indicate the main compounds of CF, red nodes denote the 9 signaling pathways screened from the KEGG analysis of the PPI network, and blue nodes symbolize the overlapped target genes linked to both CF and RA.

**Figure 4 pharmaceuticals-17-00914-f004:**
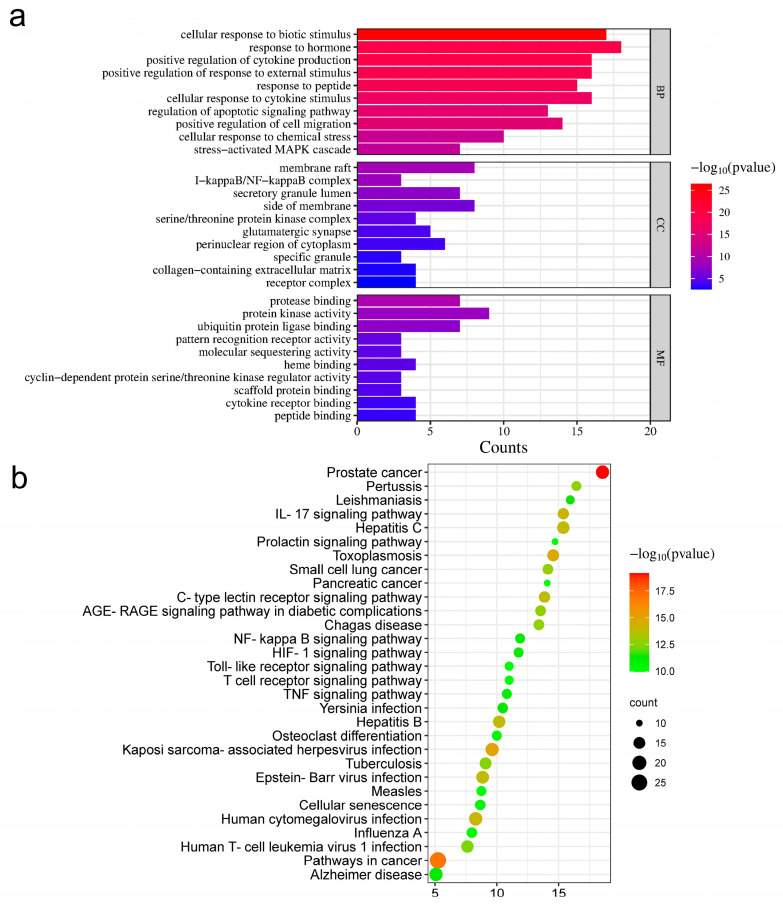
GO and KEGG functional analysis: (**a**) GO enrichment analysis. BP: GO with reference to biological processes; CC: gene ontology with reference to cellular components; MF: gene ontology with reference to molecular function; (**b**) KEGG enrichment analysis.

**Figure 5 pharmaceuticals-17-00914-f005:**
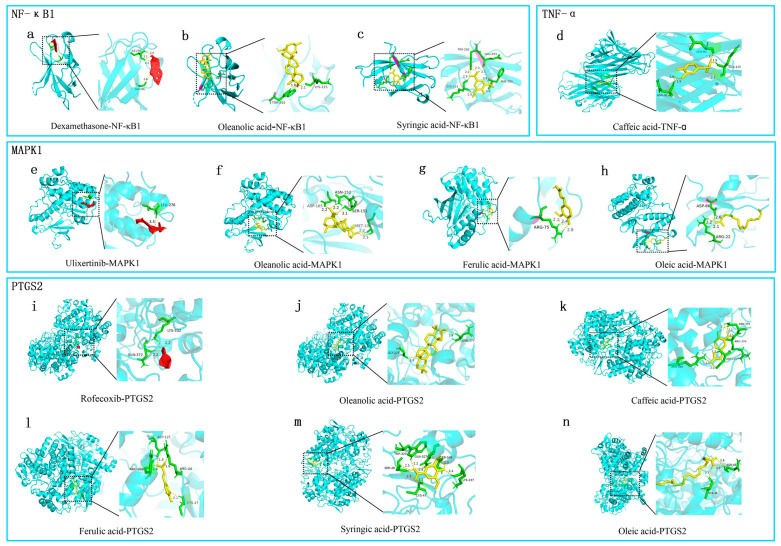
Pattern diagram of molecular docking: (**a**–**c**) molecular docking of dexamethasone, oleanolic acid, and syringic acid to NFκB1; (**d**) molecular docking of caffeic acid to TNF-α; (**e**–**h**) molecular docking of ulixertinib, oleanolic acid, ferulic acid and oleic acid to MAPK1; (**i**–**n**) molecular docking of rofecoxib, oleanolic acid, caffeic acid, ferulic acid, syringic acid, and oleic acid to PTGTS2. The yellow dashed line represents hydrogen bonds and the number next to the yellow dashed line represents the length of the hydrogen bond. The green part represents the amino acid residue. The yellow compounds were the core compounds screened from network pharmacology analysis. The red compound was inhibitor.

**Figure 6 pharmaceuticals-17-00914-f006:**
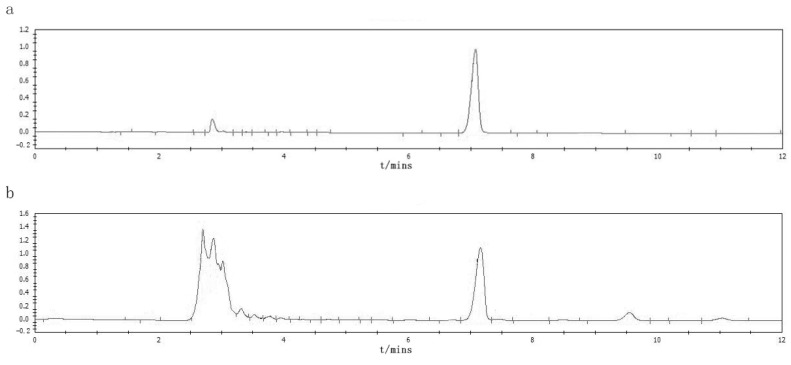
Results of HPLC: (**a**) HPLC chromatogram of the reference standard; (**b**) HPLC chromatograms of the CF sample.

**Figure 7 pharmaceuticals-17-00914-f007:**
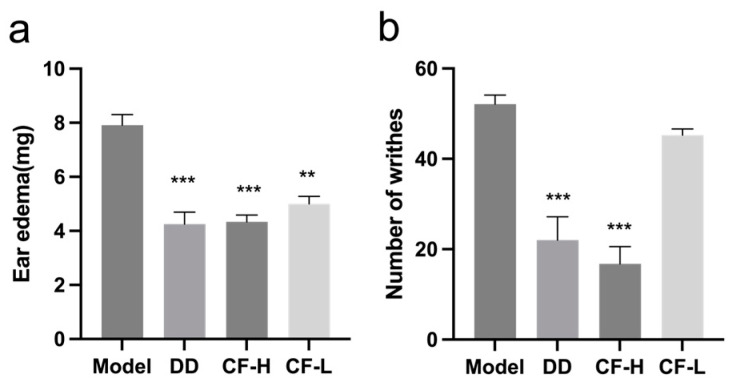
Effects of CF on inflammation and pain in mice (mean ± SEM, n = 8): (**a**) anti-inflammatory effect of CF on xylene-induced ear edema in mice; (**b**) analgesic effect of CF in the acetic acid-induced writhing model. Abbreviations: DD, diethylamine diclofenate; CF-H, CF—high dose; CF-L, CF—low dose. ** *p* < 0.01, *** *p* < 0.001 vs. model group.

**Figure 8 pharmaceuticals-17-00914-f008:**
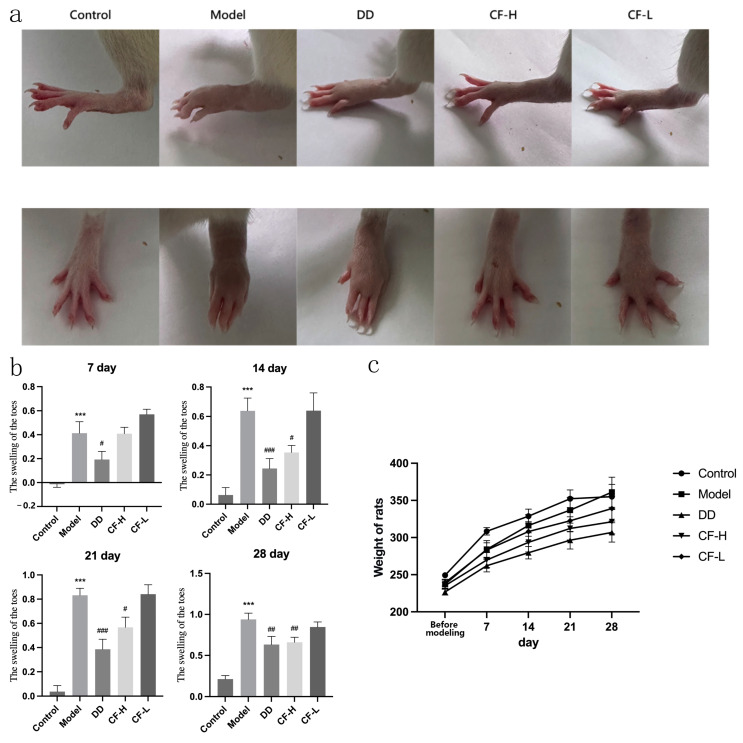
Amelioration effects of CF on arthritis in AA rats (mean ± SEM, n = 8): (**a**) macroscopic changes in arthritis of the hind limbs in rats were observed; (**b**) comparison of toe swelling in rats in each group at different times after dosage; (**c**) weight changes in rats from the induction of RA to the end of treatment. *** *p* < 0.001 vs. control group; ^#^ *p* < 0.05, ^##^ *p* < 0.01, ^###^ *p* < 0.001 vs. model group.

**Figure 9 pharmaceuticals-17-00914-f009:**
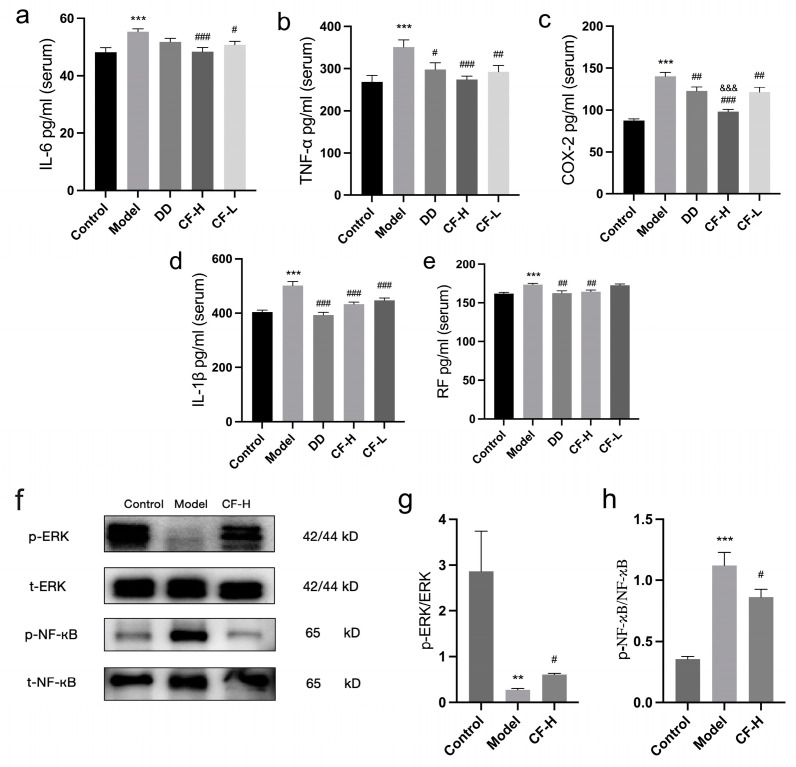
Effects of CF extracts on serum inflammation factors and protein expression in RA rats: (**a**) serum IL-6 level; (**b**) serum TNF-α level; (**c**) serum COX-2 level; (**d**) serum IL-1β level; (**e**) serum RF level (mean ± SEM, n = 8); (**f**) Western blot analysis; (**g**) effect of CF on p-ERK, ERK proteins in rats; (**h**) effect of CF on NF-κB, p-NF-κB proteins in rats (mean ± SEM, n = 5). ** *p* < 0.01, *** *p* < 0.001 vs. control group; ^#^
*p* < 0.05, ^##^ *p* < 0.01, ^###^ *p* < 0.001 vs. model group; ^&&&^
*p* < 0.001 vs. DD group.

**Figure 10 pharmaceuticals-17-00914-f010:**
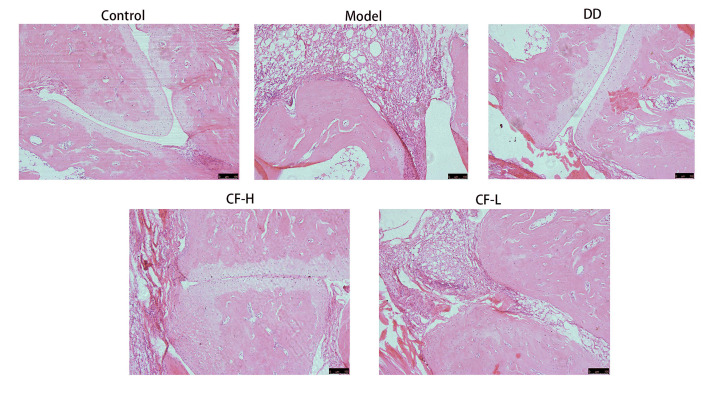
Morphology of synovial tissue in ankle joints of rats in various groups. All figures were magnified by 100×. Scare Bar: 100 μm.

**Table 1 pharmaceuticals-17-00914-t001:** Targets corresponding to inflammatory pathways.

Pathway	Target	*FDR* Values
IL-17 signaling pathway	LCN2, GSK3B, IL6, NF-κB1, TNF, PTGS2, CASP3, HSP90AA1, MAKP1, IKBKB, MAPK3, MAPK14, NF-κBIA, RELA	7.20 × 10^−15^
C-type lectin receptor signaling pathway	NF-κB1, IKBKB, NF-κBIA, RELA, PTGS2, PPP3CA, MAPK14, MAPK3, MAPK1, TNF, IL6, IL10, NF-κB2, CLEC4E	1.61 × 10^−14^
AGE-RAGE signaling pathway in diabetic complications	ICAM1, NOS3, F3, TNF, IL6, NF-κB1, SERPINE1, MAPK3, CASP3, RELA, MAPK1, CCND1, MAPK14	2.09 × 10^−13^
Human T-cell leukemia virus 1 infection	CDK2, RB1, CCND1, CDKN1A, MAPK3, PPP3CA, TP53, NFKBIA, NF-κB1, TNF, MAPK1, RELA, IL6, IKBKB, NF-κB2, ICAM1	4.24 × 10^−13^
NF-kappa B signaling pathway	PLAU, ICAM1, PTGS2, RELA, TNF, NF-κB1, NF-κB2, IKBKB, TLR4, NF-κBIA, LY96, BTK	5.96 × 10^−12^
HIF-1 signaling pathway	NOS2, NOS3, NF-κB1, INS, CDKN1A, SERPINE1, HMOX1, TLR4, MAPK3, MAPK1, RELA, IL6	6.31 × 10^−12^
TNF signaling pathway	TNF, IL6, RELA, NF-κB1, CASP3, IKBKB, PTGS2, MAPK1, MAPK3, ICAM1, MAPK14, NF-κBIA	1.51 × 10^−11^
Toll-like receptor signaling pathway	TNF, NF-κB1, LY96, TLR4, RELA, MAPK1, MAPK3, MAPK14, IL6, IKBKB, NF-κBIA	9.63 × 10^−11^
T cell receptor signaling pathway	IL10, IKBKB, TNF, NF-κB1, NF-κBIA, GSK3B, MAPK14, RELA, MAPK1, MAPK3, PPP3CA	9.63 × 10^−11^

**Table 2 pharmaceuticals-17-00914-t002:** Ranking of targets according to degree value from high to low.

Target	Full Name of Target	Target Alias	Degree
PTGS2	2Prostaglandin G/H synthase 2	COX-2	15
MAPK1	Mitogen-activated protein kinase 1	ERK2	11
NFκB1	Nuclear factor kappa-B	P50	11
TNF	Tumor necrosis factor	TNF-alpha	11
RELA	V-rel reticuloendotheliosis viral oncogene homolog A	NFκB P65	11
MAPK3	Mitogen-activated protein kinase 3	ERK1	9
PTGS1	Prostaglandin–endoperoxide synthase 1	COX-1	9
IL6	Interleukin 6	HGF	9
NFκBIA	NF-kappa-B inhibitor alpha antibody	IKBA	8
IKBKB	Inhibitor of nuclear factor kappa-B kinase subunit beta	IKK2	8
MAPK14	Recombinant human mitogen-activated protein kinase 14	CSPS	8
ADRB2	beta-2 adrenergic receptor	ADRB2R	8

**Table 3 pharmaceuticals-17-00914-t003:** The important order of core targets in the RA-related gene network.

Target	Degree	Rank
TNF	251	1/552
NFκB1	169	13/552
PTGS2	127	40/552
RELA	105	59/552
MAPK1	73	108/552

**Table 4 pharmaceuticals-17-00914-t004:** Molecular docking scores of the five core compounds and four targets.

Target	PDB ID	Compound	Affinity/(kcal·mol^−1^)	Grid Center	Grid Size
NF-κB1	**1U36**	**Dexamethasone**	**−9.17**	**(50.11, 22.21, 3.56)**	**(90, 78, 66)**
	1U36	Oleanolic acid	−6.6	(50.11, 22.21, 3.56)	(90, 78, 66)
	1U36	Syringic acid	−3.87	(50.11, 22.21, 3.56)	(90, 78, 66)
MAPK1	**4S31**	**Ulixertinib**	**−7.17**	**(−3.24, 7.31, 46.47)**	**(98, 76, 106)**
	4S31	Oleanolic acid	−8.14	(−3.24, 7.31, 46.47)	(98, 76, 106)
	4S31	Ferulic acid	−4.57	(−3.24, 7.31, 46.47)	(98, 76, 106)
	4S31	Oleic acid	−3.15	(−3.24, 7.31, 46.47)	(98, 76, 106)
PTGTS2	**5F19**	**Rofecoxib**	**−7.08**	**(22.29, 44.99, 39.17)**	**(62, 62, 84)**
	5F19	Oleanolic Acid	−10.31	(22.29, 44.99, 39.17)	(62, 62, 84)
	5F19	Caffeic acid	−5.83	(22.29, 44.99, 39.17)	(62, 62, 84)
	5F19	Syringic acid	−5.6	(22.29, 44.99, 39.17)	(62, 62, 84)
	5F19	Oleic acid	−5.27	(22.29, 44.99, 39.17)	(62, 62, 84)
	5F19	Ferulic acid	−5.2	(22.29, 44.99, 39.17)	(62, 62, 84)
TNF-α	2AZ5	Caffeic acid	−5.37	(−13.12, 69.19, 26.69)	(54, 52, 62)

## Data Availability

The datasets presented in this study are available in online repositories. All data generated or analyzed during this study can be obtained upon reasonable request to the corresponding author.

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
