# Peer review of "Exploring the Mechanism of Topical Application of Clematis Florida in the Treatment of Rheumatoid Arthritis through Network Pharmacology and Experimental Validation"

_pharmaceuticals, 2024, doi:10.3390/ph17070914_

Round 1
Reviewer 1 Report
Comments and Suggestions for Authors
The manuscript examines an experimental study on the influence of Clematis Florida (CF) in the treatment of rheumatoid arthritis. The article is well-organized, well-written, clear, and clearly defines both the objectives and the results obtained. The conclusions are adequately supported. The proposed experiments and tests are convincing and have identified a set of substances present in CF that explain why the use of CF can be beneficial as a treatment for rheumatoid arthritis.
This reviewer has only a couple of very minor comments:
- Line 54: Provide an approximate date for the publication of the book (Zhong Hua Ben Cao) and, if possible, include the original title in Chinese characters. Additionally, the transliterated title should be italicized, not enclosed in quotation marks.
- The figure captions are insufficient; please describe a bit more about what is seen in the figures in the manner of a graphical abstract.
Comments on the Quality of English LanguageThe English language used is fine.
Reviewer 2 Report
Comments and Suggestions for Authors In the current complex study, Dr. Ting Lei and colleagues explored the probable molecular mechanism of the inhibitory effect of Clematis Florida (CF) extract on rheumatoid arthritis and verified some of the obtained results in animal experiments. In particular, the authors revealed that the main metabolites of CF involved 5 compounds, including oleanolic acid, which can directly interact with various proteins, NF-kB, MAPK1, TNF-alpha and PTGS2 of which were verified by molecular docking. It should be emphasized that this study contains not only in silico data, but also their experimental verification. I expect that this paper can be published after a major revision. Major comments: -- Section 2.5 - In my opinion, an independent assessment of the involvement of the revealed core targets in the rheumatoid arthritis-associated regulome should be added to the topological analysis of the CPS network. Please reconstruct a gene network consisting of key RA-related genes downloaded from the DisGeNET database (CUI: C0003873) using STRING or GeneMANIA, embed the revealed core targets there and calculate their degree. This will allow you to rank the importance of these proteins in the pathogenesis of the disease. Otherwise, the degree given in Table 2 only indicates the number of compounds from the list of CF metabolites that can bind to the indicated protein and the involvement of this protein in the limited list of pathways listed in Figure 3c, which cannot speak to its clear association with arthritis. -- Figure 5 - In order to be more confident about the ability of the studied compounds to bind to CF-related proteins, it is necessary to (1) re-dock known inhibitors of NF-kB1, MAPK1 and PTGS2, (2) compare the location of the inhibitors with the compounds of interest in the active sites of the proteins (please add inhibitor structures colored in a different color to each docking complex shown in Figure 5) and calculate the binding energies for the inhibitors (please add these values to Table 3). Based on the molecular docking data, only the interaction of oleanolic acid with MAPK1 and PTGS2 can be considered relevant, since in other cases the free Gibbs energy was above -7 kcal/mol (the known deltaG threshold). What conclusions can you draw from this? -- Figure 9a-e - In my opinion, RA induction should cause a much more pronounced effect on the level of pro-inflammatory markers in the blood. Dear authors, what can explain such a small (less than 2-fold) increase in their levels in your experiment? Has such an increase already occurred in published data? -- Table 1 - (1) Please include p-values for each pathway; (2) Incorrect commas. Please correct; (3) It remains unclear why the authors identified a different set of terms using the same KEGG database (see Table 1 and Figure 4b). Why was it necessary to perform the KEGG analysis twice? Please explain or correct this in the text of the paper. Minor comments: Please add the chemical structures of the major CF metabolites identified by LC-MS/MS. Figure 3b,c - please enlarge the font for CF compounds to make them easier to read. Line 130 - please write p (in p-value) in italics Lines 133, 137, 138 - the phrase "CF treatment of RA" does not correspond to reality. Dear authors, you performed an enrichment analysis using PREDICTED protein targets of CF. Please correct this sentence to a more neutral one. Figure 5 - What do the dashed lines with numeric symbols indicate? H bonds? Please indicate this in the figure caption. Line 152 - please replace high with low. The lower the Gibbs free energy, the more stable the complex, the more likely the compound will interact with the protein in a living system. Figure 7 - Please decipher the abbreviations DD, CF-H, CF-L. Lines 206, 207, 209, 214 - Please correct the font of alpha, RF and beta. Line 217 - the sentence "These results verified the targets" is not correct. In ELISA, the authors only showed changes in the profile of pro-inflammatory mediators in serum in control and experimental groups, but did not prove direct interaction of CF compounds with the revealed targets. Please clarify your conclusions. Please correct. Line 280 - extra space Line 324 - (a) please add a space between 60.08 and g/kg; (b) typo in word iindicating Line 339 - please add centrifugation speed in g Line 379 - typo in word intAction Section 4.2.7 - please specify the grid box parameters used for all proteins analyzed. Line 409 and throughout the manuscript - please fill in missing spaces between the number and the unit (e.g. 10g (line 409), 0.45uM (line 410), 1000g (line 429), etc.). Line 427 - Enter the date of approval. Lines 439, 453 - typo in the word nomarl (normal?). Section 4.4.3 - please specify the dosages used as it is not clear how the high dose and low dose groups differ. Line 462 - the sentence "40 SD rats" is missing a predicate - incorrect sentence. Please correct. Line 479 - Please specify what method was used to determine foot volume?Author Response
Please see the attachment

Round 2
Reviewer 2 Report
Comments and Suggestions for Authors
I express great respect to the author's team for their serious work in improving the manuscript. All the changes and comments I noted have been duly considered and incorporated into the article. The article is ready to be published. I wish the authors success in their further research!